# TDR-HGN: Residual-enhanced heterogeneous graph networks for topology-driven feature completion

## Abstract

Heterogeneous graphs are composed of multiple types of edges and nodes. The existing heterogeneous graph neural network can be understood as a node feature smoothing process guided by the graph structure, which can accurately simulate complex relationships in the real world. However, due to real-world privacy and data scarcity, some node features are inevitably missing. Furthermore, as model depth increases and multiple types of meta-paths are aggregated, node embeddings tend to be consistent, leading to semantic confusion and overfitting problems. To improve the quality of node embeddings, we propose topology-driven residual boosting network (TDR-HGN). It introduces one-hot encoding and node type encoding to generate initial features, uses topological structure features to guide feature completion, combines residual networks to deal with semantic confusion and over-fitting problems, and builds neighbor-based high-order graph networks through meta-paths to achieve feature enhancement. We conduct extensive experiments on three heterogeneous graph datasets, and the results show that TDR-HGN can significantly improve the performance compared to other methods.

## 1 Introduction

Many real-world objects and phenomena can be accurately abstracted into network models, such as traffic networks (Xu et al., 2023) and social networks (Peng et al., 2022; Kumar et al., 2023). Network representation learning (Zhou et al., 2022) serves as the basis for downstream analysis tasks, such as node classification (Yang et al., 2022), node clustering (Wu et al., 2023), and visualization (Zhao et al., 2023a). Its goal is to learn accurate low-dimensional representations of nodes in the network. Among them, graph neural networks (GNNs) (Ju et al., 2024) are one of the most competitive network representation learning technologies, which have received extensive attention and in-depth research in academia and industry. Heterogeneous graphs contain rich semantics and can model various types of nodes and relationships. Current heterogeneous graph neural networks (HGNNs) follow a message passing framework to learn neighbor features. Although they can effectively extract multiple content and structure features, there are still two challenges that cannot be ignored.

Challenge 1: There are certain deficiencies in feature completion (Chen et al., 2020). At present, mainstream HGNN mainly uses manual methods to process nodes with missing features. When processing missing features, there are problems such as insufficient accuracy, inability to capture complex relationships, and poor versatility, resulting in the lack of depth and adaptability of the generated node representation. Recent studies have proposed using pre-trained topological learning (Jiang et al., 2021a) to guide feature completion strategies, but there are still two limitations: over-reliance on pre-training information and ignoring contextual semantic information (Dong et al., 2017; Grover & Leskovec, 2016). The pre-training information obtained based on self-supervision methods (Jiang et al., 2021b) is difficult to reflect the characteristics and task requirements of specific application scenarios, and the generated node representation has negative transfer problems and low generalization problems.

Challenge 2: Semantic confusion (Ji et al., 2021). Similar to oversmoothing in GNNs (Wang et al., 2022b), semantic confusion means that HGNN injects the semantics of multiple neighbors into

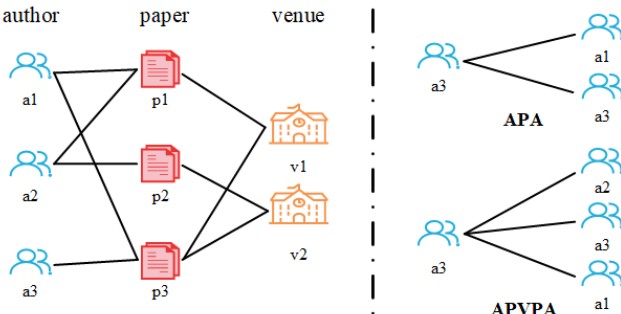

Fig. 1: An illustration of a citation network. The left figure shows the three types of nodes (i.e., author, paper, venue) and two types of connections (i.e., author-paper, paper-venue) of the citation network. The right figure shows the high-order homogeneous neighbors of the author node a3 formed by two meta-paths (i.e., Author-Paper-Author and Author-Paper-Venue-Paper-Author).

node embeddings, which makes node embeddings difficult to distinguish. This causes the model to be unable to accurately express complex relationships, seriously affecting the interpretability and visualization of the model. As can be seen from Figure 1, the first-order neighbor of node a3 is node p3. Through the meta-path APA, node a3 can obtain the features of a1 and a3. Through the meta-path APVPA, node a3 can obtain the features of a1, a2, and a3. As the number of meta-paths and the depth of the model increase, a3 aggregates features from multiple neighbors, which leads to the convergence of all node features. Second, as the length of the meta-path increases, different nodes will be connected to the same meta-path-based neighbors, and the obtained node representation is not concise but redundant.

Therefore, in addition to aggregating neighbor features, the model should also absorb the local semantics of nodes with appropriate weights. Even with multiple layers of stacking, the model can retain the underlying features of the nodes instead of injecting confusing semantic information into the node embeddings. Based on this idea, we propose a topology-driven residual enhancement network. First, by combining node category encoding and onehot encoding, we use first-order neighbors to learn topological structure features that contain more comprehensive information, use transformer-based multi-head attention mechanism and topological structure features for feature completion, capture high-order neighbors through meta-paths, and combine residual networks to retain the bottom layer of nodes. The work of this paper can be summarized as follows:

- We propose a topology-driven feature completion strategy, which introduces one-hot encoding and node type encoding to generate initial features, and uses topology to drive node feature completion.

- proposes a residual-enhanced heterogeneous graph network (TDR-HGNN), which uses a multi-head attention mechanism to capture different subspace features, and uses meta-path and residual networks to aggregate high-order neighbors and node underlying features.

- conducts extensive experiments on three datasets and compares with multiple existing methods to demonstrate the advancement and effectiveness of the method.

## 2 RELATED WORK

### 2.1 HETEROGENEOUS GRAPH NEURAL NETWORK

The HGNN method learns heterogeneous graph embedding from graph structure and node features through neural networks.According to the technology adopted in the learning mode, the existing HGNN modeling methods are mainly divided into three types: convolution-based methods, autoencoder-based methods and adversarial-based methods (Bing et al., 2023).

Convolution-based HGNN processes graph data through multiple layers of stacked heterogeneous convolutional layers. Meta-path-based models capture high-order neighborhood features by aggre-

gating nodes at both ends of meta-paths. Wang et al. (2019) designs hierarchical attention model HAN calculates the importance scores of nodes and meta-paths through node-level and semantic-level attention, and learns node embeddings in heterogeneous graphs based on the scores. MAGNN (Fu et al., 2020) uses linear mapping to solve the dimension mismatch problem and adopts a single-layer GCN to aggregate all nodes in the meta-path. The relation-based model selects the most useful meta-path for downstream tasks by comparing the importance between nodes of different types, providing support for the interpretability of the model. A model with a hierarchical aggregation architecture (Yang et al., 2021) is proposed to dynamically assign edge weights through improved graph convolution kernels and attention mechanisms to accurately aggregate neighbor feature. To enhance the robustness of node embedding, HGSL (Zhao et al., 2021) uses graph attention network and multi-view learning to capture the latent relations in the graph structure. AC-HEN (Wang et al., 2022a) generates multi-view embeddings through feature aggregation and structure aggregation, and fills in missing features with the embedding fusion module in the weakly supervised learning paradigm. Meta-path-based methods ignore first-order neighbors and edge types, resulting in insufficient learning feature in HG. Relation-based methods require more model parameters to handle complex interactions between multiple types of nodes and edges, increasing space consumption.

Autoencoder-based methods reduce the dimension of input feature through encoders and restore data to high-dimensional space through decoders, and learn effective embeddings by minimizing the reconstruction error between input data and restored data. Wang et al. (2021) proposes a heterogeneous graph attention autoencoder HGATE, which reconstructs node features and edges of heterogeneous graphs through stacked encoder and decoder layers, and captures semantic information by combining node-level and semantic-level attention. Liu et al. (2023) proposes a new bidirectional encoding HGBER unsupervised framework, which discovers the optimal node distribution by introducing the minimization of the clustering constraint objective function L3, making the representation of nodes of different categories more dense and compact. Adversarial methods (Lan et al., 2020; Zhao et al., 2020) utilize the competition mechanism between the generator and the discriminator of the generative adversarial network to improve the quality and diversity of node and edge generation. Autoencoder-based methods require a lot of computing resources and are overly dependent on the original graph structure. Adversarial methods focus on utilizing graph structure feature and ignore content features. Therefore, comprehensive considerations are needed to design efficient and accurate models.

## 2.2 FEATURE COMPLETION IN GRAPH NEURAL NETWORKS

In the real world, due to privacy or other reasons, relationships are partially observed, resulting in incomplete graph structures and missing node features. To learn feature completion in incomplete heterogeneous graphs, HGNN-AC (Jin et al., 2021) uses existing heterogeneous network embedding methods to obtain the topological structure embedding of nodes, guides the weighted aggregation of neighbor node features based on the topological structure feature, and uses the MAGNN (Fu et al., 2020) model to further enhance node embeddings. AC-HEN (Wang et al., 2022a) adopts a multi-view fusion strategy to capture richer feature representations through three views. It uses the k-nearest neighbor method to select similar nodes in the feature space, uses GCN to aggregate neighbor nodes in the structure space, and combines random walk sampling to aggregate high-order neighbors.

Li et al. (2023) proposes a heterogeneous residual graph attention network HetReGAT-FC, and designed the HetReGAT module through onehot encoding and multi-head attention mechanism. Similar to the idea of HGNN-AC, it uses the HetReGAT module to learn the topological structure feature of the heterogeneous graph, and uses the attention coefficient obtained from the topological feature as a guide for feature completion. HOAE (Li et al., 2024) uses a self-attention mechanism based on high-level transformers to fill in missing features and uses first-order neighbors to enhance node embeddings. RA-HGNN (Zhao et al., 2023b) introduces a type conversion matrix to optimize the embedding of heterogeneous network graphs and uses a residual attention network for feature completion. At the same time, Xia Yong's random dropout method reversely optimizes the feature completion performance. However, HGNN-AC, RA-HGNN, and HetReGAT-FC do not consider high-order neighbors of the same type resulting in suboptimal node embeddings. AC-HEN uses stacked GCN layers for multi-neighbor feature aggregation, which easily leads to over-smoothing problems. HOAE performs feature aggregation based on multiple GAT layers, ignoring the information in the feature space.

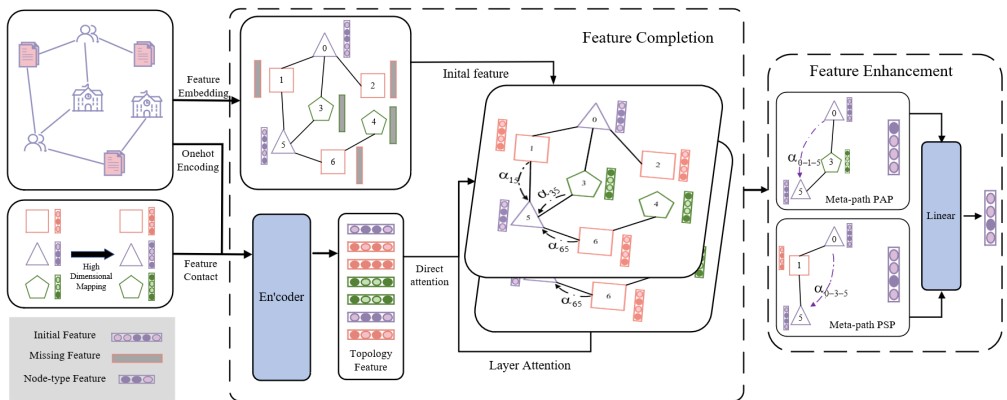

Fig. 2: A brief illustration of the overall framework of TDR-HGNN

## 3 PRELIMINARY

**Definition 1: Heterogeneous Graph (HG).** A heterogeneous graph is represented as $\mathcal{G} = (V, E, T, R)$, which consists of nodes $V$ and the corresponding node type set $T$, and edges $E$ and the corresponding edge-type set $R$. In a heterogeneous graph, $|T|$ and $|R|$ represent the number of node types and the number of edge types respectively, $|T| + |R| > 2$. $A \in \mathcal{R}^{N \times N}$ denotes the adjacency matrix and $N$ denotes the number of nodes.

**Definition 2: Meta-path.** A meta-path can be defined as a specific path : $v_1 \xrightarrow{r_1} v_2 \xrightarrow{r_2} \cdots \xrightarrow{r_l} v_{l+1}$, where $v \in T$ and $r \in R$. It describes a complex relationship $r = r_1 \circ r_2 \circ \cdots \circ r_l$ between node $v_1$ and node $v_{l+1}$, where $\circ$ represents a combinatorial operation between relations.

## 4 METHODOLOGY

In this section, we introduce the overall framework(in Fig.2) of TDR-HGNN in detail.

### 4.1 FEATURE COMPLETION OF RESIDUAL ATTENTION MECHANISM BASED ON TOPOLOGICAL STRUCTURE

The network homogeneity principle states that similar nodes are more likely to form connections, which affects the formation and evolution of the network. Due to the noise and incompleteness of the initial graph nodes, using the original features to obtain the attention coefficient cannot accurately reflect the relationship between nodes when the features are not rich. Using the topological structure to guide message passing can effectively capture the overall connection pattern and network structure, and reduce the impact of feature sparsity and noise on the attention mechanism. First, TDR-HGNN uses the node onehot encoding $X^{onehot} \in R^{N \times N}$ and the node type encoding $X^{type} \in R^{N \times |T|}$ to obtain the model input $H^{in}$:

$$H^{in} = (X^{onehot}W_1 + b_1)||(X^{type}W_2 + b_2) \tag{1}$$

where $H^{in} \in R^{N \times 2d_{model}}$ represents the input of the model, $d_{model}$ and $N$ represent the the dimension of the hidden layer and number respectively, and $||$ represents the concatenation operation, $W_1, W_2, b_1$ and $b_2$ are all trainable neural weights and biases of the linear transformation. Then, an attention mechanism is used to guide the nodes to aggregate global structural feature:

$$e_{v,u}^{src} = h_v^{in}W_{src}$$
$$e_{v,u}^{dst} = h_u^{in}W_{dst} \tag{2}$$

where $e_{v,u}^{src}$ represents the attention coefficient starting from node $v$, and $e_{v,u}^{dst}$ represents the attention coefficient ending at node $u$, $W_{src}$ and $W_{dst}$ are all trainable neural weights and biases of the linear transformation, $h_v^{in}$ and $h_v^{in}$ represent feature vectors of nodes $v$ and $u$ in $H^{in}$. $e_{v,u}$ is the sum of

the attention coefficients of the starting point and the end point:

$$e_{v,u} = e_{v,u}^{src} + e_{v,u}^{dst} \tag{3}$$

The softmax function is used to normalize the attention scores of multiple edges of a single node to eliminate the dimension effect.

$$\alpha_{v,u} = \frac{\exp(e_{v,u})}{\sum_{n \in \mathcal{N}(v)} \exp(e_{v,n})} \tag{4}$$

The calculated normalized weights (in Eq.4) are used to guide the aggregation of neighbor features to obtain the topological feature $h_v$ of node $v$.

$$h_v^l = \sigma \left( \sum_{n \in \mathcal{N}_v} \alpha_{v,n}^{(l)} \cdot h_n^{l-1} \right) \tag{5}$$

where $\sigma$ represent activate function. Faced with the differences in data dimensions and the complexity of data content, TDR-HGNN introduces a multi-head attention mechanism to process input sequences in parallel from different angles, improving the model's ability to understand and capture complex dependencies:

$$h_v^l = \|_{k=1}^K \sigma \left( \sum_{n \in \mathcal{N}_v} \left[ \alpha_{v,n}^l \right]_k \cdot h_n^{l-1} \right) \tag{6}$$

where $l$ represents the layer of the model. The dimension of $H_l$ is transformed from $N \times d_{model}$ to $N \times (K * d_{model})$ because of the multi-head attention mechanism, so TDR-HGNN introduces a linear transformation to transform the output dimension of layer $l$ to $N \times d_{model}$:

$$H^l = H^l W^{topo} + b^{topo} \tag{7}$$

where $H^0 = H^{in}$ in Eq.1, the output of the last layer $H^l$ is used as the topological structure feature $H^{topo}$ for feature completion. TDR-HGNN enriches the topological structure feature $H^{topo}$ by aggregating high-order neighbor features through multi-layer propagation, ensuring that the attention coefficient $\beta_{v,u}$ calculated by $H^{topo}$ can more accurately reflect the relationship between nodes.

$$\beta_{v,u} = \frac{\exp(W_\beta h_u^{topo})}{\sum_{n \in \mathcal{N}_v} \exp((W_\beta h_n^{topo})} \tag{8}$$

where $h_u^{topo}$ and $h_n^{topo}$ represent features of nodes $v$ and $n$ in $H^{topo}$, $\beta_{v,u}$ represents the attention coefficient between node $v$ and node $u$. $\beta$ is used to guide the aggregation of the model input $h^{in}$.

$$h_v = \sigma \left( \sum_{n \in \mathcal{N}_v} \beta_{v,n} \cdot h_n^{in} \right) \tag{9}$$

where $h_v^{in} = (X_v^{in})^M (X_v^{onehot})^{1-M}$, $X^{in}$ represents the initial features of the node, $M$ represent combination coefficient. Since multi-layer models are prone to overfitting and semantic confusion, TDR-HGNN uses a residual connection mechanism to allow nodes to adaptively retain features:

$$h_v^l = \sigma \left[ \sum_{n \in \mathcal{N}_v} \beta_{v,n}^l h_n^l + W_{res}^l h_v^{l-1} \right] \tag{10}$$

where $W_{com}$ and $W_{res}$ denotes parameter matrices for node completion and residuals. For the attention coefficients of different layers, TDR-HGNN also connect through residual network. The specific form is as follows:

$$\beta_{v,u}^l = (1-\eta)\hat{\beta}_{v,u}^l + \eta\beta_{v,u}^{l-1} \tag{11}$$

Similarly, multi-head attention mechanism is introduced to capture the multivariate relationships between nodes:

$$h_v^l = \|_{k=1}^K \sigma \left( \sum_{n \in \mathcal{N}_v} \left[ \beta_{v,n}^l \right]_k \cdot h_n^{l-1} \right) \tag{12}$$

The parameter matrix $W_{com}$ is used for dimension conversion:

$$H^l = H^l W_{com} \tag{13}$$

The output $H^l$ of the last layer is used as the node embedding after feature completion, which is used for subsequent feature enhancement. Algorithm 1 shows the process of feature completion.

## 4.2 META-PATH-BASED HIGH-LEVEL FEATURE ENHANCEMENT

---
**Algorithm 1** Feature Completion Algorithm

---
**Require:** The heterogeneous graph $\mathcal{G} = (V, E, T, R)$, the initial features $X^{in}$, the node type feature $X^{type}$, the node onehot encoding vector $X^{onehot}$, the residual layer hyperparameters $\eta$ and the number of attention head $K$.

**Ensure:** the completed feature $h$

1: Initialize the input of the topology feature module $H^{in}$ through Eq.(1)
2: **for** $k = 1$ to $K$ **do**
3:     **for** $v \in V$ **do**
4:         Find the node neighbors $\mathcal{N}_v$
5:         Calculate the edge attention coefficient $e_{v,u}$ through Eq.(2,3)
6:         **for** $u \in \mathcal{N}_v$ **do**
7:             Calculate the node attention coefficient $\alpha_{v,u}$ through Eq.(4)
8:         **end for**
9:         Calculate the node $v$ embedding $H_v^l$ through Eq.(5)
10:     **end for**
11:     Concatenate all embedding $H$ from all attention head through Eq.(6)
12: **end for**
13: Transforme the dimension of the topological feature $H^{topo}$ through Eq.(7)
14: **for** $k = 1$ to $K$ **do**
15:     **for** $v \in \mathcal{N}_v$ **do**
16:         Calculate the node attention coefficient $\beta_{v,u}^l$ through Eq.(8)
17:         **if** $l > 1$ **then**
18:             Residual connection attention coefficient $\beta_{v,u}^l$ through Eq.(11)
19:             Residual connection the node $v$ embedding $h_v^l$ through Eq.(10)
20:         **end if**
21:         Calculate the node $v$ embedding $h_v^l$ through Eq.(9)
22:     **end for**
23:     Concatenate all embedding $h_v$ from all attention head through Eq.(12)
24: **end for**
25: Transforme the dimension of the node feature $h_v$ through Eq.(13)
26: Return $h$

---

In order to aggregate high-order homogeneous neighbor features, TDR-HGNN uses meta-paths to divide HG into multiple isomorphic subgraphs. Given multiple meta-paths $P = \{P_1, \cdots, P_m\}$, the normalized attention coefficient $\gamma_{v,u}^{p_i}$ of node $v$ under meta-path $P_i$ is defined as follows:

$$\gamma_{v,u}^{p_i} = \frac{\exp\left(\sigma\left(h_v W_{p_i} h_u\right)\right)}{\sum_{n \in \mathcal{N}_v} \exp\left(\sigma\left(h_v W_{p_i} h_n\right)\right)}. \tag{14}$$

where $W_{p_i}$ is the learnable parameter matrix under the meta-path $P_i$. Similarly, multi-path aggregation introduces a multi-head attention mechanism to stabilize the data variance during node aggregation:

$$z_v^{p_i} = \|_{k=1}^K \sigma\left(\sum_{u \in N_v^{p_i}} \left[\gamma_{v,u}^{p_i}\right]_k \cdot h_u\right), \tag{15}$$

Different meta-paths represent different semantic information. TDR-HGNN needs to set different weight coefficients to balance the feature of multiple meta-paths:

$$s_{p_i} = \frac{1}{|T|} \sum_{v \in T} \sigma\left(W_z \cdot z_v^{p_i} + b_z\right) \tag{16}$$

$$e_{p_i} = q^T \cdot s_{p_i} \qquad (17)$$

where $q^T$ is a learnable attention weight vector, multiple meta-paths share the parameter $q^T$, $W_z$ and $b_z$ are learnable parameter matrices, $\sigma$ indicates non-linear activation functions.

$$\gamma_{p_i} = \frac{\exp\left(e_{p_i}\right)}{\Sigma_{p_j \in P} \exp\left(e_{p_j}\right)}, \qquad (18)$$

$$z_v = \sum_{p_i \in P} \gamma_{p_i} \cdot z_v^{p_i}, \qquad (19)$$

Finally, we pass the linear encoder to get the final node, which functions similarly to a linear classifier:

$$O_v = \sigma\left(W_o \cdot z_v\right) \qquad (20)$$

where $\sigma(\cdot)$ indicates non -linear activation functions. $W_o$ is a learning weight parameter.

## 4.3 MODEL EVALUATION

Our model is applied to the semi-supervised classification task by defining the cross entropy loss function as the optimization function:

$$loss = -\sum_{v \in y_L} Y_{v_l} \cdot \ln\left(C \cdot O_{v_l}\right), \qquad (21)$$

where $Y_{v_i}$ and $O_{v_l}$ represent the category and predicted probability of node $v_i$ respectively, $y_L$ denotes the set of labeled nodes, $C$ represents the parameters of classifier. REHG-TAC optimizes the parameters by minimizing the Eq.(21) using the gradient descent method.

## 5 EXPERIMENTS

### 5.1 IMPLEMENTATION DETAILS

This paper uses three different heterogeneous graph datasets for experiments to comprehensively evaluate the performance of the model in processing heterogeneous graphs. We compare TDR-HGNN with ten competitors, among which HAN (Wang et al., 2019), GTN (Yun et al., 2019), MAGNN (Fu et al., 2020), HGSL (Zhao et al., 2021), RoHE (Zhang et al., 2022) and HetReGAT-FC (Li et al., 2023) are convolution-based methods, ie-HG (Yang et al., 2021), AC-HEN (Wang et al., 2022a), RA-HGNN (Zhao et al., 2023b) and HOAE (Li et al., 2024) are encoder-based methods.

We use the same dataset partitioning ratio and meta-path for all models, and take the average results of five experiments as the final test results. For different models, we select the model layer with the best performance on the validation set as the baseline to ensure that each model can maximize the advantages of its structure to achieve the best performance. In the experiment, the hidden layer dimension is set to 64, the output layer dimension is set to 16, the dropout rate is set to 0.5, the weight decay is set to 0.001, and the number of attention heads K is set to 8, because 8 attention heads can produce more stable results. For AC-HEN and RA-HGNN, we use MAGNN as the downstream model.

### 5.2 NODE CLASSIFICATION

We use SVM for node classification with training rates ranging from 20% to 80%. ACM and IMDB are node classification datasets with original features, while DBLP is a node classification dataset without original features. According to Table1, TDR-HGNN, HOOE, HRG-FC, and RA-HGNN generally outperform HAN, GTN, MAG, IEHG, and ROHE. This is because the feature completion module can help the model obtain richer node representations and use structure feature to alleviate the sparsity problem of node features. Compared with RA-HGNN, TDR-HGNN has an overall improvement of 0.6%-3.0%, which may be because RA-HGNN only relies on original feature for feature completion, and the expression ability of nodes is limited. Compared with HRG-FC, TDR-HGNN has an overall improvement of 0.2%-1.0%, which may be because HRG-FC adopts a relationship-based approach, which leads to certain limitations in capturing high-order neighbor functions. Compared with HOAE, TDR-HGNN has an overall improvement of 0.6%-1.0%, which may be due to the overfitting and semantic confusion caused by the multi-layer stacking of HOAE.

Table 1: Performance (%) of TDR-HGNN and other models on the task of node classification(the best results are highlighted in bold).

| Dataset | Metrics | Ratio | Without Feature Completion | | | | | | With Feature Completion | | | | |
|---|---|---|---|---|---|---|---|---|---|---|---|---|---|
| | | | HAN | GTN | MAG | ie-HG | HGSL | RoHE | AC-HEN | RA-HGNN | HRG-FC | HOAE | TDR-HGNN |
| ACM | Ma-F1 | 20% | 90.71 | 90.88 | 88.20 | 91.35 | 92.43 | 91.57 | 90.61 | 90.55 | 92.63 | 93.11 | **93.80** |
| | | 40% | 91.33 | 91.36 | 89.60 | 92.14 | 92.58 | 92.01 | 91.12 | 91.13 | 93.53 | 93.54 | **94.11** |
| | | 60% | 91.73 | 91.74 | 90.48 | 92.59 | 92.73 | 92.34 | 91.62 | 91.52 | 93.82 | 93.79 | **94.50** |
| | | 80% | 91.91 | 91.81 | 90.89 | 92.79 | 92.83 | 92.50 | 92.11 | 91.96 | 93.89 | 93.83 | **94.72** |
| | Mi-F1 | 20% | 90.59 | 90.76 | 88.28 | 91.27 | 92.38 | 91.47 | 90.75 | 89.41 | 92.54 | 93.03 | **93.71** |
| | | 40% | 91.22 | 91.24 | 89.70 | 92.11 | 92.54 | 91.94 | 91.19 | 90.70 | 93.45 | 93.47 | **94.02** |
| | | 60% | 91.60 | 91.61 | 90.51 | 92.53 | 92.69 | 92.25 | 91.71 | 91.55 | 93.73 | 93.69 | **94.39** |
| | | 80% | 91.76 | 91.70 | 90.91 | 92.73 | 92.77 | 92.38 | 92.16 | 91.98 | 93.87 | 93.73 | **94.62** |
| DBLP | Ma-F1 | 20% | 92.63 | 93.99 | 92.93 | 92.73 | 93.72 | 92.39 | 92.51 | 93.62 | 93.87 | 93.32 | **93.88** |
| | | 40% | 92.87 | 94.27 | 93.32 | 93.57 | 93.65 | 92.77 | 93.24 | 93.89 | 94.00 | 93.87 | **94.21** |
| | | 60% | 93.05 | 94.15 | 93.69 | 93.66 | 93.81 | 92.84 | 93.69 | 94.08 | 94.11 | 94.02 | **94.38** |
| | | 80% | 93.16 | 94.26 | 94.01 | 94.09 | 94.09 | 93.11 | 93.81 | 94.27 | 94.32 | 94.21 | **94.64** |
| | Mi-F1 | 20% | 93.20 | 94.45 | 93.45 | 93.24 | 94.19 | 92.90 | 93.05 | 94.02 | 94.07 | 93.89 | **94.33** |
| | | 40% | 93.43 | 94.71 | 93.82 | 94.00 | 94.09 | 93.28 | 93.74 | 94.26 | 94.32 | 94.10 | **94.64** |
| | | 60% | 93.61 | 94.60 | 94.18 | 94.10 | 94.23 | 93.34 | 94.19 | 94.44 | 94.57 | 94.48 | **94.81** |
| | | 80% | 93.69 | 94.70 | 94.48 | 94.47 | 94.52 | 93.55 | 94.29 | 94.71 | 94.88 | 94.67 | **95.04** |
| IMDB | Ma-F1 | 20% | 58.11 | 57.26 | 57.87 | 58.24 | 58.09 | 57.76 | 58.45 | 58.26 | 59.38 | **59.39** | 59.37 |
| | | 40% | 58.56 | 57.90 | 59.23 | 59.33 | 58.24 | 57.93 | 59.46 | 59.46 | 60.01 | 59.67 | **60.11** |
| | | 60% | 58.73 | 58.04 | 59.72 | 59.65 | 58.80 | 58.04 | 59.83 | 59.95 | 60.44 | 59.80 | **60.50** |
| | | 80% | 58.88 | 58.84 | 59.94 | 59.87 | 58.93 | 58.13 | 59.78 | 60.19 | 60.67 | 59.95 | **60.79** |
| | Mi-F1 | 20% | 58.14 | 57.12 | 57.89 | 58.16 | 58.44 | 58.03 | 58.17 | 58.27 | 59.32 | **59.68** | 59.56 |
| | | 40% | 58.58 | 57.81 | 59.29 | 59.26 | 58.56 | 58.21 | 59.45 | 59.49 | 60.13 | 59.99 | **60.35** |
| | | 60% | 58.72 | 57.89 | 59.80 | 59.57 | 58.98 | 58.32 | 59.97 | 59.98 | 60.55 | 60.13 | **60.71** |
| | | 80% | 58.91 | 58.74 | 60.06 | 59.82 | 59.09 | 58.41 | 60.03 | 60.24 | 60.76 | 60.26 | **61.00** |

## 5.3 NODE CLUSTERING

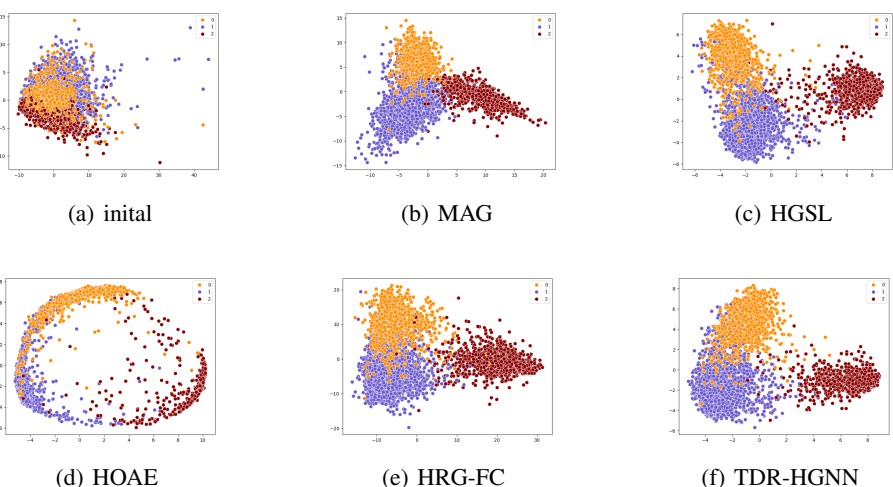

(a) inital     (b) MAG     (c) HGSL

(d) HOAE     (e) HRG-FC     (f) TDR-HGNN

Fig. 3: Visual representation of the training results of the ACM dataset.

In the visualization experiment on the ACM dataset, we use principal component analysis (Maćkiewicz & Ratajczak, 1993) to project node embeddings into two-dimensional space. Fig.3 shows that the clustering distance of the model MAG without feature completion is smaller, indicating that feature completion can improve node embedding quality. In contrast, although the inter-cluster distance of HOAE is larger, the points within the cluster are more scattered, the cluster distance of the HRG-FC model is closer, and the clustering performance of HGSL and TDR-HGNN is better.

Table 2: Test results of four feature completion modules on the ACM dataset (MAGNN as a downstream model).

| Metrics(%) | Ratio | MAG$_{avg}$ | MAG$_{HRG-FC}$ | MAG$_{HOAE}$ | MAG$_{TDR-HGNN}$ |
|---|---|---|---|---|---|
| Macro-F1 | 20% | 87.82 | 91.86 | 92.97 | **93.71** |
| | 40% | 89.39 | 92.41 | 93.26 | **94.00** |
| | 60% | 90.35 | 93.61 | 93.66 | **94.41** |
| | 80% | 90.81 | 93.85 | 93.81 | **94.50** |
| Micro-F1 | 20% | 87.91 | 91.83 | 92.86 | **93.62** |
| | 40% | 89.42 | 92.68 | 93.13 | **93.92** |
| | 60% | 90.37 | 93.74 | 93.56 | **94.31** |
| | 80% | 90.83 | 93.35 | 93.77 | **94.39** |

Table 3: Test results of four feature completion modules on the ACM dataset (ie-HG as a downstream model).

| Metrics(%) | Ratio | ie-HG$_{avg}$ | ie-HG$_{HRG-FC}$ | MAG$_{HOAE}$ | ie-HG$_{TDR-HGNN}$ |
|---|---|---|---|---|---|
| Macro-F1 | 20% | 91.24 | 91.35 | 91.44 | **91.75** |
| | 40% | 92.12 | 92.26 | 91.96 | **92.62** |
| | 60% | 92.27 | 92.48 | 92.32 | **92.91** |
| | 80% | 92.38 | 93.14 | 92.86 | **93.21** |
| Micro-F1 | 20% | 91.14 | 91.42 | 91.57 | **91.63** |
| | 40% | 92.03 | 92.11 | 92.01 | **92.56** |
| | 60% | 92.17 | 92.43 | 92.29 | **92.78** |
| | 80% | 92.26 | 92.97 | 92.79 | **93.01** |

## 5.4 COMPATIBILITY

In order to evaluate the compatibility of different feature completion methods with heterogeneous graph models, we selected two mainstream heterogeneous graph models, MAGNN and ie-HGCN, for node classification tasks. Four different feature completion methods, avg, HetReGAT-FC, HOAE, and TDR-HGNN, were used in the experiments on the ACM dataset.

The results show that the average interpolation strategy weakens the quality of embedding and performs poorly in downstream tasks. Compared with MAG$_{HRG-FC}$ and MAG$_{HOAE}$, the performance of our feature completion model is improved by nearly 0.5% to 1.5%. Compared with ie-HG$_{HRG-FC}$ and ie-HG$_{HOAE}$, ie-HG$_{TDR-HGNN}$ also has an improvement of nearly 0.5%. This is because the HRG-FC model uses topological feature to complete feature completion, focusing on the connection relationship between nodes, while ignoring the subtle differences between node features. The HOAE model takes into account the local contextual feature of nodes, but cannot fully capture the complex patterns of graph structures. The results show that the feature completion module in the TDR-HGNN model can adapt well to most heterogeneous graph neural networks and show significant compatibility.

## 5.5 ABLATION EXPERIMENTS

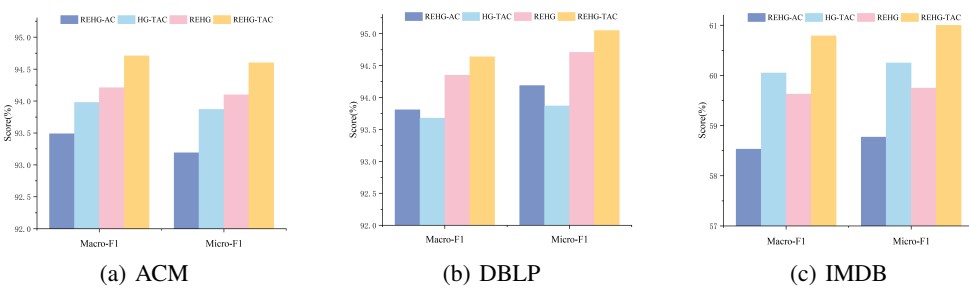

(a) ACM      (b) DBLP      (c) IMDB

Fig. 4: Experimental study of ablation of TDR-HGNN.

In order to effectively evaluate the effectiveness of each component in TDR-HGNN separately, we designed three TDR-HGNN variants for ablation studies:

- REHG-AC: It is a variant of TDR-HGNN that obtains attention coefficients from the original graph for feature propagation, instead of guiding the feature aggregation of the original graph by calculating topological structure feature.
- HG-TAC: It is a variant of TDR-HGNN that eliminates the residual connections in feature completion.
- REHG: It is a variant of TDR-HGNN, which eliminates the node category encoding and uses the original features as the input of the completion module.

We conducted node classification experiments on three variants of TDR-HGNN and presented classification results with a training rate of 80% on three datasets. On the IMDB dataset with a more complex graph structure, the performance of REHG-AC is significantly lower than that of TDR-HGNN, which shows that the relationship between nodes can be effectively captured using graph topology feature. Compared with HG-TAC, TDR-HGNN performance improved by 0.48% to 0.75%, which shows that residual connections help transfer the underlying features of nodes and alleviate the over-smoothing problem caused by model stacking. Compared with REHG, TDR-HGNN performance improved by 0.3% to 2.75%, indicating that node type encoding, as important prior information, can help the model understand and utilize the complex structural feature of heterogeneous graphs.

## 5.6 HYPER-PARAMETER ANALYSIS

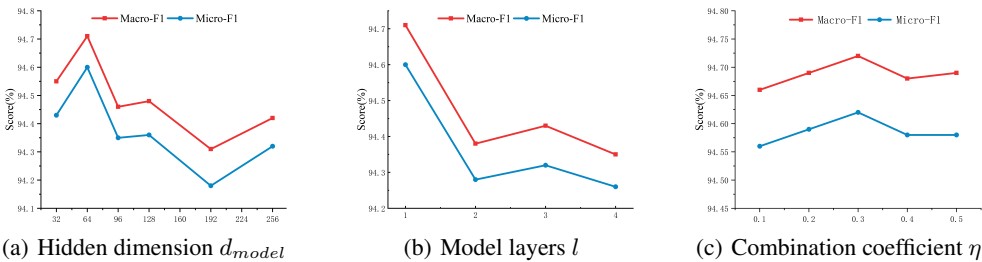

(a) Hidden dimension $d_{model}$     (b) Model layers $l$     (c) Combination coefficient $\eta$

Fig. 5: Analysis of hyperparameters (Hidden dimension $d_{model}$, Model layers $l$, Combination coefficient $\eta$) on the ACM dataset

We tested the impact of three hyperparameter values on model performance on the ACM dataset, and the score of each hyperparameter is the average result calculated based on 80% of the training ratio. As can be seen from Fig.5, when the initial hidden layer dimension increases, the model performance improves, but too large a hidden layer dimension causes the model to overfit the training data, thereby reducing the generalization ability to new data. In addition, the increase in the number of initial layers is conducive to the model learning higher-level abstract features, but too many model layers will produce gradient vanishing or gradient explosion, leading to semantic confusion and overfitting problems. The combination coefficient $\eta$(in Eq.11) reflects the degree of dependence of parameters between multiple layers. When $\eta$ is too large, it may make it difficult for the model to learn new feature during training and ignore new patterns that may exist in the data.

## 6 CONSLUSION AND FUTURE WORK

This paper proposes a residual enhanced heterogeneous graph network for topology-driven feature completion. It uses the topological structure feature and meta-paths of HGNN for feature completion and enhancement, and combines the residual network and the advanced attention mechanism based on Transformer to guide message passing. In the future, we will consider exploring more intrinsic relationships between multiple meta-paths and generalize the TDR-HGNN framework to dynamic heterogeneous graphs.

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

# A APPENDIX

## A.1 PRELIMINARY

**Problem 1: Heterogeneous Graph Embedding.** For a given heterogeneous graph $\mathcal{G}$, the goal of node embedding is to learn a mapping function $f : V \to \mathcal{R}^d : v \to h$, where $d \ll |V|$. The function $f$ aims to accurately reflect the connection relationship between nodes.

**Problem 2: Feature Completion.** Given a heterogeneous graph $\mathcal{G} = (V, E, T, R)$ as input, the goal of feature completion is to learn a mapping function $f_1(\mathcal{G}, A, h) : h \to h^+$. $h$ denotes the node with missing features, and $h^+$ denotes the node with completed features. The function $f_1$ aims to make $h^+$ close to real features.

**Problem 3: feature Enhancement.** Given a heterogeneous graph $\mathcal{G} = (V, E, T, R)$ as input, feature enhancement aims to learn a mapping function $f_2(\mathcal{G}, A, h) : h \to z$, where $z$ denotes the enhanced node feature. Through function $f_2$, the node feature $z$ can learn the neighbor features and global structural feature.

## A.2 ALGORITHM

Algorithm 2 shows the main process of feature enhancement.

---
**Algorithm 2** feature Enhancement Algorithm

---
**Require:** The heterogeneous graph $\mathcal{G} = (V, E, T, R)$, the completed node feature $h$, the multiple meta-paths $P = \{P_1, \cdots, P_m\}$ and the number of attention head $K$.
**Ensure:** The final embedding $z$.
 1: Initialize the input of the topology feature module $H^{in}$ through Eq.(1)
 2: **for** $P_i \in P$ **do**
 3:     **for** $v \in V$ **do**
 4:         Find the node neighbors $\mathcal{N}_v$
 5:         **for** $u \in \mathcal{N}_v$ **do**
 6:             Calculate the node attention coefficient $\gamma_{v,u}^{p_i}$ through Eq.(14)
 7:         **end for**
 8:         Calculate the node $v$ embedding $z_v$ through Eq.(15)
 9:     **end for**
10:     Concatenate all embedding $z_v^{p_i}$ from all attention head through Eq.(15)
11:     Transform the dimension of the node embedding through Eq.(16)
12:     Calculate the normalized meta-path coefficient $\gamma_{p_i}$ through Eq.(17,18)
13: **end for**
14: Aggregating embeddings from multiple meta-paths through Eq.(19)
15: Return $z$

---

## A.3 DATASET

This paper uses three different heterogeneous graph datasets for experiments to comprehensively evaluate the performance of the model in processing heterogeneous graphs. The relevant information of the three datasets is shown in Table 4.

- ACM: It is a subset of the ACM dataset, which is a citation network containing 4,019 papers ($P$), 7,167 authors ($A$), and 60 topics ($S$). The features of the papers in the dataset are bags of keywords, which are divided into three categories: database, wireless communication, and data mining according to the labels. In the experiment, we choose two types of meta-paths $\{PSP, PAP\}$.

- DBLP: It is a subset of the DBLP dataset, which is an academic network containing 4,057 authors (A), 14,328 papers (P), 8,789 terms (T), and 20 positions (V). The features of the authors in the dataset are bags of keywords. In the experiment, we use three types of meta-paths: $\{APA, APTPA, APVPA\}$.

Table 4: Details of three datasets.

| Dataset | Nodes | Target node | Arrributes | Meta-path |
|---------|-------|-------------|------------|-----------|
| ACM | paper: 4019
author: 7167
subject: 60 | paper | paper: original
auther: missing
subject: missing | PSP
PAP |
| DBLP | auther: 4057
paper: 14328
term: 8789
venue: 20 | auther | auther: missing
paper: original
term: missing
venue: missing | APA
APTPA
APVPA |
| IMDB | movie: 4278
actor: 5257
director: 2081 | movie | movie: original
actor: missing
director: missing | MAM
MDM |

- IMDB: It is the IMDB dataset itself, which is a movie dataset containing 4,278 movies ($M$), 5,257 actors ($A$), and 2,081 directors ($D$). The features of the movies in the dataset are bags of keywords, and the labels are divided into three categories. In the experiment, we choose two types of meta-paths: $\{MAM, MDM\}$.

## A.4 BASLINE

We compare REHG-TAC with ten competitors and the details are as follows:

- HAN (Wang et al., 2019): A hierarchical attention model that calculates the importance scores of nodes and meta-paths through node-level and semantic-level attention, and learns node embeddings in heterogeneous graphs based on the scores
- GTN (Yun et al., 2019): It uses Graph Transformer Layer to automatically learn useful meta-paths and multi-hop connections, and generates new meta-path graphs to achieve effective node representation learning.
- MAGNN (Fu et al., 2020): It optimizes HAN and uses all node features on meta-paths to achieve more powerful heterogeneous graph representation learning. It is referred to as MAG in subsequent experiments.
- ie-HG (Yang et al., 2021): It decomposes HG into multiple bipartite graphs, and uses node-level aggregation and semantic-level aggregation to assign different weights to each bipartite graph to capture relationship information. It is referred to as ie-HG in subsequent experiments.
- HGSL (Zhao et al., 2021): It uses graph attention network and multi-view learning to capture the potential relationship in the graph structure, effectively improving the flexibility and accuracy of embedding extraction
- RoHE (Zhang et al., 2022): It equips with attention purifier to mask the noise information of topological attack to improve the robustness of the model.
- AC-HEN (Wang et al., 2022a): It generates multi-view embeddings through feature aggregation and structure aggregation, and combines the embedding fusion module in the weakly supervised learning paradigm for feature completion.
- RA-HGNN (Zhao et al., 2023b): It completes the features of missing nodes through the topological structure of heterogeneous graphs and residual networks, and enhances node embeddings using the completed embeddings and MAGNN model.
- HetReGAT-FC (Li et al., 2023): It is designed through one-hot encoding and multi-head attention mechanism. Similar to the idea of HGNN-AC, it uses the HetReGAT module to learn the topological structure feature of heterogeneous graphs and uses the attention coefficient obtained from the topological feature as a guide for feature completion. It is referred to as HRG-FC in subsequent experiments.
- HOAE (Li et al., 2024): It completes the missing features of nodes through the self-attention mechanism based on advanced Transformer and combines meta-path to learn high-order neighbor features.

Table 5: Performance (%) of REHG-TAC and other models on the task of node clustering(the best results are highlighted in bold).

| Datasets | ACM | | DBLP | | IMDB | |
|---|---|---|---|---|---|---|
| | NMI | ARI | NMI | ARI | NMI | ARI |
| HAN | 68.61 | 71.62 | 65.92 | 67.37 | 12.98 | 13.46 |
| MAG | 70.16 | 72.14 | 78.67 | 84.02 | 13.08 | 12.76 |
| ie-HG | 49.47 | 34.89 | 32.33 | 27.21 | 13.08 | 13.04 |
| HGSL | 72.25 | 76.25 | 77.63 | 82.47 | 6.21 | 8.78 |
| ROHE | 69.21 | 72.48 | 70.84 | 77.26 | 12.39 | 12.89 |
| AC-HEN | 70.45 | 73.88 | 77.19 | 82.05 | 9.32 | 10.18 |
| RA-HGNN | 65.21 | 69.88 | 79.59 | 84.92 | **14.31** | 14.12 |
| HRG-FC | 73.85 | 78.19 | 80.33 | 86.72 | 13.94 | 14.11 |
| HOAE | 73.68 | 77.36 | 79.64 | 85.11 | 12.20 | 10.52 |
| REHG-TAC | **77.05** | **81.63** | **81.14** | **86.35** | 14.16 | **14.57** |

## A.5 NODE CLUSTERING

The NMI and ARI evaluation indicators in Table 5 show that the REHG-TAC model has a performance improvement of nearly 4% 10% compared with the model without feature completion on ACM and DBLP. Compared with HOAE, the NMI and ARI of HRG-FC and REHG-TAC have an improvement of nearly 1.5% 3%, indicating that the topological feature of the node has a positive impact on clustering. In addition, we found that residual networks can effectively improve the node clustering effect (RA-HGNN and REHG-TAC have higher scores on the IMDB dataset).

