# OpenReview forum: "TDR-HGN:Residual-enhanced heterogeneous graph networks for topology-driven feature completion"
_ICLR.cc/2025/Conference — Submitted to ICLR 2025_

### Official Review · Reviewer_vLoA · 2024-10-31

**Soundness:** 2
**Presentation:** 2
**Contribution:** 2
**Rating:** 3
**Confidence:** 4

**Summary:**

The paper introduces a Topology-Driven Residual-Enhanced Heterogeneous Graph Network (TDR-HGN) consisting of node type and one-hot encodings to generate initial features, residual networks to mitigate over-fitting and semantic confusion, and high-order neighbor aggregation through meta-paths to achieve feature enhancement. The main aim is to improve the feature completion in heterogeneous graphs and to prevent embeddings from becoming overly uniform, which leads to loss of semantic information.

**Strengths:**

1. The topic is interesting and relevant.
2. propose a different architecture for heterogeneous graphs along with feature completion
3. The proposed model demonstrates improvement over selected baselines.
4. authors provided comprehensive results across multiple tasks, including node classification and clustering, with ablation studies and hyperparameter analysis.

**Weaknesses:**

1. While the model outperforms the selected baselines,  the authors should include a wider range of recent and diverse approaches.
2. Small datasets have been chosen. authors should conduct experiments on larger datasets and provide a discussion on the scalability of the model for very large graphs.
3. No insights have been provided into how the proposed model performs on computational efficiency/training time (compared to baselines).
4. One-hot encoding is infeasible for large-scale graphs. How do you plan to handle large graphs?
5. Sections of the methodology (e.g. the notation and details in Sections 4.1 and 4.2) could be improved for better understanding.
6. the authors should share the code with pseudocode.
7. There are numerous typos, and the paper requires proper refinement. For example, in eq. 4, alpha should be written alpha^l.
8. the proposed architecture is not very well motivated and the novelty of its contributions is somewhat unclear.

**Questions:**

1. Are there specific models you believe should be included in the baseline comparisons, especially in the context of heterogeneous/heterogeneous multiplex/knowledge graphs?
2. Do you have plans to make the code available publicly?
3. Since the model is evaluated on a single dataset, can you discuss the potential challenges in applying your model to other datasets?
4. Could you further clarify the novelty of the added components and explain how these modifications specifically address the unique challenges of heterogeneous multiplex graphs?

**Details Of Ethics Concerns:**

See the "weaknesses"

---

### Official Review · Reviewer_J8Uv · 2024-11-03

**Soundness:** 2
**Presentation:** 3
**Contribution:** 2
**Rating:** 3
**Confidence:** 3

**Summary:**

The paper presents the topology-driven residual boosting network (TDR-HGN) for heterogeneous graphs, addressing key challenges such as missing node features, semantic confusion, and overfitting in existing graph neural networks (GNNs). The proposed approach integrates one-hot and node type encoding to enhance initial feature generation and employs topological structures for feature completion. It also incorporates residual networks to mitigate confusion and overfitting, alongside constructing high-order graph networks through meta-paths. The extensive experimental evaluation on three heterogeneous graph datasets demonstrates significant performance improvements over current methodologies, highlighting the effectiveness and potential of TDR-HGN in enhancing node embedding quality.

**Strengths:**

1. The paper compares against relatively new baseline methods, and the experimental results are impressive, enhancing the credibility and applicability of the research.
2. The proposed method is easy to follow, with a clear overall structure that facilitates reader comprehension and potential application.

**Weaknesses:**

1. The method appears to be more about feature engineering, designing multiple features with prior knowledge to input into the heterogeneous graph attention network, potentially lacking deeper model innovation.
2. Using one-hot encoding for certain features may lead to memory issues in large datasets, and the paper does not analyze the time or space complexity of the approach.
3. The paper does not explore whether applying the used feature information to more advanced heterogeneous graph network algorithms could yield better performance.
4. While ablation studies indicate that residual connections and topology-based attention have a significant impact on performance, the importance of the initial one-hot features and other features remains unclear.

**Questions:**

1. Have the authors considered testing the proposed features within more advanced heterogeneous graph network algorithms, and what insights could this provide regarding the robustness of their method?
2. In the ablation studies, can the authors provide a more detailed analysis of the significance of the initial one-hot features and other feature types in relation to the performance outcomes?

---

### Official Review · Reviewer_619R · 2024-11-04

**Soundness:** 2
**Presentation:** 3
**Contribution:** 2
**Rating:** 3
**Confidence:** 3

**Summary:**

The paper proposes a feature completion model for heterogeneous graphs that alleviates semantic confusion and overfitting for long meta-paths. It introduces one-hot encoding to generate initial features and uses topological structure features to guide feature completion. Meanwhile, it combines residual networks to deal with semantic confusion and over-fitting problems to enrich the generated features with high-order neighbors. The experimental results show improvement.

**Strengths:**

1. It proposes a feature completion model that incorporates high-order neighbors to enrich the generated features.
2. It combines both structural embeddings and existing features of other nodes for missing nodes.
3. it uses attention modules to model the importance of neighbors with only one-hot embedding and type embedding.

**Weaknesses:**

1. It's not reasonable to me that only one node embedding to calculate attention coefficients for all nodes starting from node $v$ and all nodes ending with node $u$ in Eq. (2), and then add them up to get the attention coefficient for an edge in Eq. (3) . Since the one-hot encoding and type embedding are both very sparse and $e_{v,u}^{src}$ or  $e_{v,u}^{dst}$ are identical to every neighbor of node $v$ or $u$. I doubt it can learn a good coefficient that can distinguish the importance.
2. some typos. e.g., two $h_v^{in}$ at line 215, Transforme at line 294
3. A similar issue also exists in Eq. (8), with no $h_v$ involved in the attention calculation.
4. The experimental results are not significant.

**Questions:**

1. Do you include $h_v^{l-1}$ in Eq. (5) as well?
2. Can you also explain why choosing weights as the residual target in Eq. (11)?

---

### Official Review · Reviewer_7Xh7 · 2024-11-07

**Soundness:** 2
**Presentation:** 2
**Contribution:** 2
**Rating:** 3
**Confidence:** 3

**Summary:**

This paper proposes a topology-driven residual enhancement network (TDR-HGN) for heterogeneous graph neural networks (HGNNs) to address the challenges of feature completion and semantic confusion. The TDR-HGN introduces one-hot encoding and node type encoding to generate initial features and uses topological structure features to guide feature completion. The proposed model combines residual networks to deal with semantic confusion and overfitting problems, and builds neighbor-based high-order graph networks through meta-paths to achieve feature enhancement. Extensive experiments on three heterogeneous graph datasets demonstrate that TDR-HGN significantly outperforms some existing methods.

**Strengths:**

1. The paper introduces a topology-driven feature completion strategy that leverages one-hot encoding and node type encoding to generate initial features. This approach is new in its emphasis on using topological structure features to guide feature completion.
2. The methodology is well-defined and logically structured. Further, this paper conducts experiments on several data sets to verify its effectiveness.

**Weaknesses:**

1. The novelty is trivial. The residual network structure and the feature completion method are off-the-shelf ideas.
2. Some importance reference is missing, such as [1]. Compared with ref.[1], the proposed method achieves few improvements.
3. Minor writing issues: the sentences between line 92-96 lack subjects.

[1] Du C, Yao K, Zhu H, et al. Seq-HGNN: learning sequential node representation on heterogeneous graph[C]//Proceedings of the 46th International ACM SIGIR Conference on Research and Development in Information Retrieval. 2023: 1721-1730.

**Questions:**

None.

---

### Meta-Review · Area_Chair_Tasy · 2024-12-21

**Metareview:**

The paper proposes a topology-driven residual enhancement network for heterogeneous graph neural networks. The topic is important and popular. The model design is reasonable. Experiments show that the proposed method outperforms baseline models. Many issues need to be addressed, such as limited novelty, missing important references, unclear method design, and writing problems. Reviewers are negative about this work.

**Additional Comments On Reviewer Discussion:**

No discussion is necessary as all give rejection.

---

### Decision · Program_Chairs · 2025-01-22

Reject